# Encoding of multi-modal emotional information via personalized skin-integrated wireless facial interface

Jin Pyo Lee[1,2], Hanhyeok Jang[1], Yeonwoo Jang [1], Hyeonseo Song[1], Suwoo Lee[1], Pooi See Lee [2] ✉ & Jiyun Kim [1,3] ✉

Human affects such as emotions, moods, feelings are increasingly being considered as key parameter to enhance the interaction of human with diverse machines and systems. However, their intrinsically abstract and ambiguous nature make it challenging to accurately extract and exploit the emotional information. Here, we develop a multi-modal human emotion recognition system which can efficiently utilize comprehensive emotional information by combining verbal and non-verbal expression data. This system is composed of personalized skin-integrated facial interface (PSiFI) system that is self-powered, facile, stretchable, transparent, featuring a first bidirectional triboelectric strain and vibration sensor enabling us to sense and combine the verbal and non-verbal expression data for the first time. It is fully integrated with a data processing circuit for wireless data transfer allowing real-time emotion recognition to be performed. With the help of machine learning, various human emotion recognition tasks are done accurately in real time even while wearing mask and demonstrated digital concierge application in VR environment.

The utilization of human affects, encompassing emotions, moods, and feelings, is increasingly recognized as a crucial factor in improving the interaction between humans and diverse machines and systems[1–3]. Consequently, there is a growing expectation that technologies capable of detecting and recognizing emotions will contribute to advancements across multiple domains, including HMI device[4–6], robotics[7–9], marketing[10–12], healthcare[13–15], education[16–18], etc. By discerning personal preferences and delivering immersive interaction experiences, these technologies have the potential to offer more user-friendly and customized services. Nonetheless, decoding and encoding emotional information poses significant challenges due to the inherent abstraction, complexity, and personalized nature of emotions[19,20]. To overcome these challenges, the successful utilization of comprehensive emotional information necessitates the extraction of meaningful patterns through the detection and processing of combined data from multiple modalities, such as speech, facial expression, gesture, and various physiological signals (e.g., temperature, electrodermal activity)[21–23]. Encoding these extracted patterns into interaction parameters tailored for specific applications also becomes essential.

Conventional approaches for recognizing emotional information from humans often rely on analyzing images of facial expressions[24–26] or speech of verbal expression[27–29]. However, these methods are frequently impeded by environmental factors such as lighting conditions, noise interference, and physical obstructions. As an alternative, text analysis techniques[30–32] have been explored for emotion detection, utilizing vast amounts of information available on diverse social media platforms. However, this approach presents challenges due to the diverse ambiguities and new terminologies being introduced, which further complicates the accurate detection of emotions from the text.

[1]School of Material Science and Engineering, Ulsan National Institute of Science and Technology, Ulsan 44919, South Korea. [2]School of Materials Science and Engineering, Nanyang Technological University, 50 Nanyang Avenue, Singapore 639798, Singapore. [3]Center for Multidimensional Programmable Matter, Ulsan National Institute of Science and Technology, Ulsan 44919, South Korea. ✉e-mail: pslee@ntu.edu.sg; jiyunkim@unist.ac.kr

To overcome these limitations, sensing devices capable of capturing changes in physiological signals, including EEG[33–35], EMG[36–38], ECG[39–41] and GSR[42–44] have been employed to collect more accurate and reliable data. These devices can establish correlations between these signals and human emotions irrespective of environmental factors, but the requirement of bulky equipment limits their application to everyday communication scenarios.

In recent studies, flexible skin-integrated devices have shown the possibility of providing real-time detection and recognition of emotional information through various modalities such as facial expressions, speech, text, hand gestures, physiological signals, etc.[45–56]. Specifically, a resistive strain sensor has been employed to directly detect facial strain deformations that occur during facial expressions[46,47,51,52]. This approach offers simplicity by using thin and soft skin-integrated electrode interfaces for current flow, allowing for wearable or portable applications. However, an additional power source, low working frequency range, and extra components for the signal conversion cause simple modality only limited to one-to-one correlation that imposes constraints on the range of applications such as healthcare, VR where complementary information is needed to approximate natural interaction, and user experience can be enhanced by multiple ways of inputs. Furthermore, most existing studies have primarily focused on recognizing and exploiting human emotions, intentions or commands using the single-modal data that can have weaknesses in specific context, thus limiting the use of higher-level and comprehensive emotional contexts[45,48–50,53–56]. On the other hand, to overcome the drawbacks of each modality for a more resilient system, multi-modal emotion recognition was conducted to draw embedded high-level information by using the combined knowledge from all the accessible data sensing[57–59]. Consequently, to effectively and precisely encode emotional information, an advanced format of the skin-integrated device necessitates improved wearability seamlessly integrating with individuals, while possessing multi-modal sensing capabilities to process and extract higher-level of information. Also, this personalized device, capable of real-time collection of reliable and accurate multi-modal data regardless of external environmental factors, should be accompanied by the corresponding classification technique to encode the gathered data into personalized feedback parameters for target applications.

Here, we proposed a human emotion recognition system in an attempt to utilize complex emotional states with our personalized skin-integrated facial interface (PSiFI) offering simultaneous detection and integration of facial expression and vocal speech. The PSiFI incorporates a personalized facial mask that is self-powered, easily applicable, stretchable, transparent, capable of wireless communication, and highly customized to conformally fit into an individual's face curvatures based on 3D face reconstruction. These features enhance the device's usability and reliability in capturing and analyzing emotional cues, facilitating the real-time detection of multi-modal sensing signals derived from facial strains and vocal vibrations. To encode the combinatorial sensing signals into personalized feedback parameters, we employ a convolutional neural network (CNN)-based classification technique that rapidly adapts to an individual's context through transfer learning. In the context of human emotion recognition, we specifically focus on facial expression and vocal speech as the chosen multi-modal data, considering their convenience for data collection and classification based on prior research findings.

The PSiFI device is basically comprised of strain and vibration sensing units based on triboelectrification to detect facial strain for facial expression and vocal vibration for speech recognition, respectively. The incorporation of a triboelectric nanogenerator (TENG) enables the sensor device to possess self-powering capabilities while offering a broad range of design possibilities in terms of materials and architectures[60,61], thus fulfilling the requirements of personalized and multi-modal sensing devices. The sensing units are made of PDMS film as a dielectric layer and PEDOT:PSS coated PDMS film as an electrode layer prepared by the semi-curing method which enables the film to exhibit good transparency with decent electrical conductivity. Furthermore, we demonstrated real-time emotion recognition with data processing circuit for wireless data transfer and real-time classification based on rapidly adapting convolution neural network (CNN) model with the help of transfer learning using data augmentation methods. Last, we demonstrated digital concierge application as an exciting possibility in virtual reality (VR) environment via human machine interfaces (HMIs) with our PSiFI. The digital concierge recognizes a user's intention and interactively offers helpful services depending on the user's affectivity. Our work presents a promising way to help to consistently collect data regarding emotional speech with barrier-free communication and can pave the way toward acceleration of digital transformation.

## Results

### Personalized skin-integrated facial interface (PSiFI) system

We devised personalized skin-integrated facial interface (PSiFI) system consisting of multimodal triboelectric sensors (TES), data processing circuit for wireless data transfer and deep-learned classifier. Figure 1A illustrates the schematics of overall process for human emotion recognition with PSiFI from fabrication to classification task. As for making personalized device, we brought in 3D face reconstruction process by collecting 3D data of user's appearance from scanned photos and converting the data to digital models. This process allowed us to fabricate personalized device fitted in well with various user faces and successfully secure individual user data for accurate recognition task. (Supplementary Fig. 1). Subsequently, we utilized both verbal/non-verbal expression information detected from multimodal sensors and classified human emotions in real-time using transfer learning applied convolution neural network (CNN).

As shown in Fig.1B, the emotional information based on verbal/non-verbal expression in the form of digital signals was sent to be the PSiFI mask and wirelessly transferred with data processing circuit. To effectively detect the signals for the emotional information, the PSiFI was integrated with multi-modal TES to capture facial skin strains and vocal cord vibrations by detecting electrical signals from glabella, eye, nose, lip, chin and vocal cord selected as representative regions based on previous studies regarding facial muscle activation patterns during facial expression[62–64].

Figure 1C provides the schematic and real image of the TES consisting of simple two-layer structure where PEDOT:PSS-coated polydimethylsiloxane (PDMS) and nanostructured PDMS were used as stretchable electrode and dielectric layer respectively so that our TES are based on single electrode mode in principle. Figure 1D shows schematics of the PEDOT:PSS-coated PDMS and dielectric layers for each strain and vibration type. The PEDOT:PSS-coated PDMS was fabricated by semi-cured process[65,66] where coating is conducted before full-curing of the elastomer (Supplementary Movie 1). Our stretchable electrode based on the semi-curing process was characterized and showed better performance when it compared to conventional surface treated electrode in terms of optical, mechanical, and electrical aspects. (Supplementary Fig. 2) As shown in scanning electron microscope (SEM) image in Fig.1D, for the dielectric layers we fabricated, nano surface engineering was introduced by inductively coupled plasma reactive ion etching process (ICP-RIE) to improve triboelectric performance by enhancing specific surface area. (Supplementary Fig. 3) Additionally, the dielectric layer for the vibration sensing was perforated like the acoustic holes which enhance vibrate the volume of air inside (Supplementary Movie 2).

### Working mechanism and characterization of the strain sensing unit

Converting facial skin strain during facial expression into distinct electrical signals and sending the data as non-verbal information to the

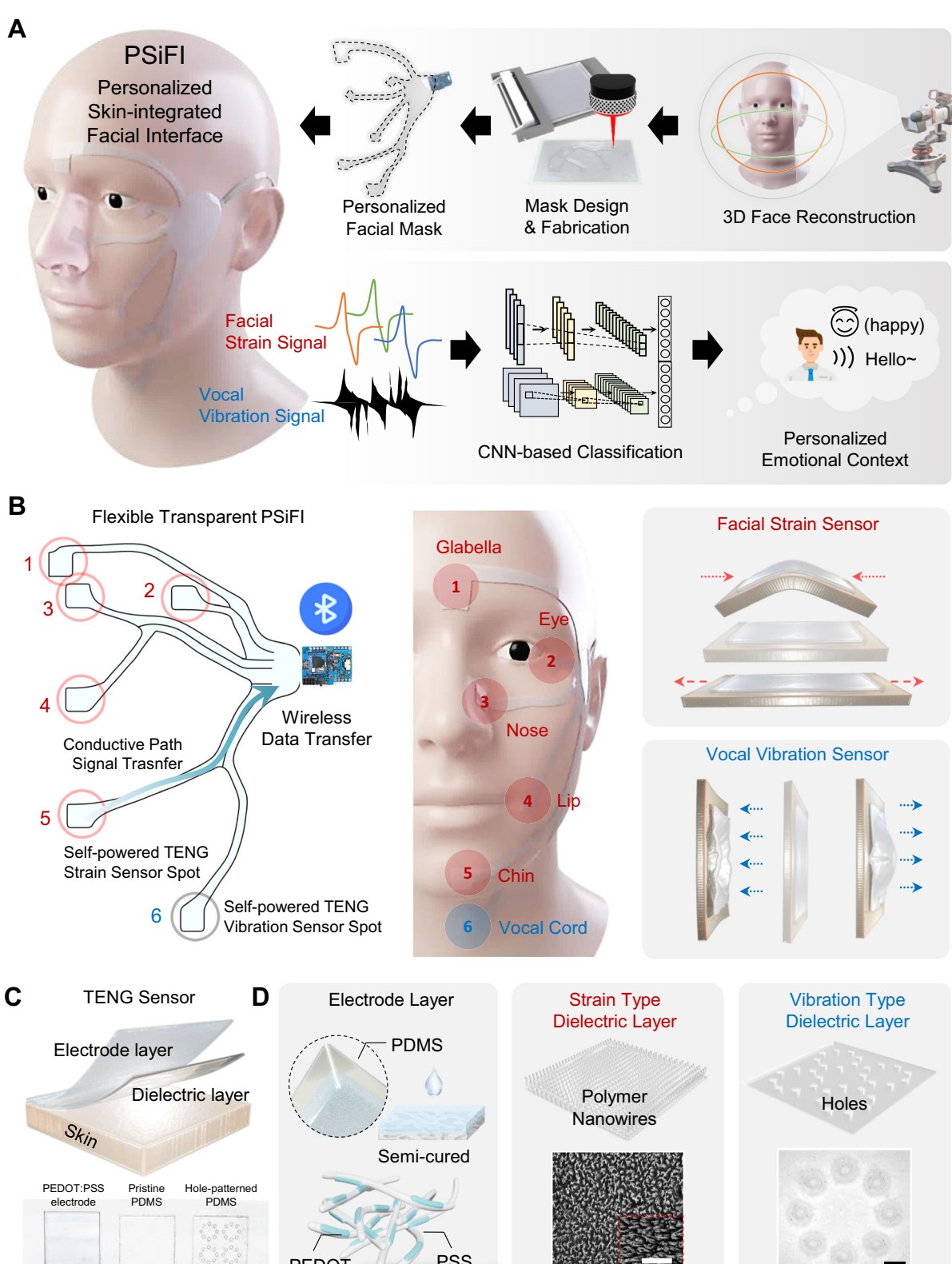

circuit system is the function of our strain sensing unit. As depicted schematically in Fig. 2A, the strain sensing unit was fabricated with the nanostructured PDMS for its high effective contact area as a dielectric layer and PEDOT:PSS embedded PDMS as an electrode layer to make TES with the single electrode structure for simple configuration to be facilitated as wearable sensors. These two layers were separated by double sided tapes applied to both ends of the layers as a spacer to be

consistently generate a series of electrical signals during the operation cycle. Besides, all the parts in the sensing units are made of stretchable and skin-friendly viable materials and can be prepared through scalable fabrication processes (for the details see the "Methods" section and Supplementary Fig. 4). These characteristics of the materials used in the strain sensing unit allow our strain sensor to retain relatively good electrical conductivity even under stretching in the range of

**Fig. 1 | The system overview with PSiFI. A** Schematic illustration of personalized skin-integrated facial interfaces (PSiFI) including triboelectric sensors (TES), data processing circuit for wireless communication and deep-learned classifier for facial expression and voice recognition. **B** Schemes showing 2d layout for the PSiFI in the form of wearable mask and depicting two different types of TES in terms of sensory stimulus such as facial strain and vocal vibration. **C** Schematic diagram of the TES which consists of simple two-layer structure such as electrode layer and dielectric layer and photograph of the TES components, respectively. Scale bar: 1 cm.

**D** Schematics demonstrating fabricated components for our TES. As for the electrode layer, PEDOT:PSS based electrode was made via semi-curing process. (left). As for the dielectric layer, it was designed differently considering sensing stimuli such as strain and vibration to achieve optimal sensing performance. The inset in center showing SEM image for nanostructured surface of strain type dielectric layer and in right showing photograph for punched holes as acoustic holes of vibration type dielectric layer. Scale bar: 2 μm and 1 mm.

facial skin strain during facial expression and guarantee robustness of the sensing unit. As schematically shown in Fig. 2B, an electrical potential builds up due to the difference between triboelectric series based on different affinity for electrons, which the PDMS played a triboelectrically negative material by receiving electrons and the PEDOT:PSS based stretchable electrode played a triboelectrically positive material by donating electrons in TES. On top of that, our strain sensing unit makes the contact area changes when stretched and achieved even buckled states so that it can detect bidirectional strain motion among the triboelectric based strain sensors for the first time, according to our knowledge. Correspondingly, the generated output signals of our strain sensing unit during the buckle-stretch cycle were shown in Fig. 2C. The comprehensive working mechanism of the bidirectional strain sensor for each mode was demonstrated in Supplementary Fig. 5.

To characterize the strain sensing unit in terms of mechanical and electrical properties, a linear motor was employed to exert a cyclic force on the sensing unit as shown in Fig. 2D. Figure 2E and F provides our strain sensing unit sensitivity measurement in a strain range from 0% to 100% by buckling and stretching, respectively. The sensitivity was derived from $S = \Delta V/\Delta \varepsilon$ where $\Delta V$ is the relative potential change and $\varepsilon$ is the strain. As for the buckling strain, linearity of the electrical responses and a sensitivity of 5 mV was obtained in a strain range up to 50% despite non-linear region occurred beyond the strain due to anomalous shape change. The signals in the non-linear region were differentiated with the difference in the width of time as shown in Supplementary Fig. 6. As for the stretching strain, an acceptable linearity and sensitivity of 3 mV was obtained in wide strain range up to 90%. We measured the response time of the strain sensing unit to evaluate the performance of the unit as it can be executed real-time classification tasks. As shown in Fig. 2G, there is no apparent latency time between the stretching force and corresponding the output voltage so that we can make sure the sensing unit can detect the sensing in real time. The stretch–release of one cycle (Fig. 2G, inset) exhibits a response time of below 20 ms. Therefore, compared with other strain sensors, our strain sensing unit has an advantage because of its high sensitivity in bi-direction, fast-response time and high stretchability, which can ensure an accurate sensing of the facial expression via converted electrical signals in real time.

We also measured the output voltage at constant strain of 40% depending on the working frequencies ranging from 0.5 to 3 Hz and confirmed that our strain sensing unit can show reliable performance regardless of the frequencies as shown in Fig. 2H. When it comes to long-term use in practical application, the mechanical stability of our sensing unit also can be considered as important property. As demonstrated in Fig. 2I, apparent output voltages changes were not observed for the strain sensing unit after 3000 continuous working cycles under 40% strain. It is noteworthy that the 40% strain change is way beyond the requirement for most facial skin strain during facial expression demonstrations[45,67].

### Working mechanism and characterization of the vocal sensing unit

Our vocal sensing unit has a function of capturing vocal vibrations on the vocal cord during verbal expression and sending the data as verbal

information to the circuit system. As shown in Fig. 3A, the vocal sensing unit was fabricated with the holes patterned PDMS as dielectric layer and PEDOT:PSS embedded PDMS as an electrode layer to make TES. The holes were introduced into the vocal sensing unit as acoustic holes which not only act as communicating vessels to ventilate an air between two contact surfaces to the ambient air, which results in enhanced flat frequency response but also reduce the stiffness by improving the movement of the rim of diaphragms[68–70] (Supplementary Fig. 7 and Table S1). To be configured into TES, like the strain sensing unit, the dielectric and electrode layer were separated by double-sided tapes applied to both ends of the layers as a spacer for consistent operations during working cycles. The inset to Fig. 3A provides an enlarged view of the vocal sensing unit capturing vocal vibrations on vocal cord. As schematically depicted in Fig. 3B, an electrical potential builds up due to triboelectric series difference based on an electron affinity. Figure 3C provides the schematic drawing showing hole pattern configuration applied in vocal vibration sensor to see how the pattern influence the output and SEM images of the holes.

We measured output voltage signals of the vibration sensing units with different open ratios (ORs) considered the proportion of area perforated with acoustic holes in the whole area on the frequency response of the devices as shown in Fig. 3D. The frequency ranges we tested encompass the fundamental frequency of typical adult men and women ranging from 100 to 150 Hz (Fig. 3D, blue) and from 200 to 250 Hz (Fig. 3D, red), respectively[71]. The results indicate that the vibration sensing unit with OR value of 10 exhibited best output voltage performance and the wideset bandwidth of flat frequency response. This experimental observation is originated from a trade-off between the deflection of dielectric layer and the effective contact area. Larger OR leads to a larger deflection of the dielectric diaphragm and thus a higher electric output. However, increased OR will reduce the effective contact area for triboelectrification, and thus a lower electrical output. Accordingly, an optimized value of OR is needed for maximization of the electrical output. Figure 3E provides measured data plots of output voltage signals per each different OR at the testing frequency of 100 Hz.

As shown in Fig. 3F and G, the output voltage of the vibration sensing unit was affected by structural parameters such as the support thickness and number of holes. As the support thickness is increased, the gap between the triboelectric layers is larger so that the effective contact area can be reduced thus the generated triboelectric output signals is decreased. On the other hand, the larger number of holes with the same OR condition makes the diaphragms deflect more vigorously, thus enhancing the triboelectric output performance. These experiments were carried out at the testing frequency of 100 Hz. Lastly, as shown in Fig. 3H, we measured the output voltage between the vibration sensing unit with and without holes as a function of input vibration acceleration in the ranging from 0.1 to 1.0 g at the same testing frequency of 100 Hz. Both sensing units have a uniform sensitivity obtained from dividing the measured output voltage by the vibration acceleration. As for the sensitivity, the hole-patterned vibration sensing units exhibits 5.78 V/g around 2.8 times larger than that of the pristine vibration sensing unit.

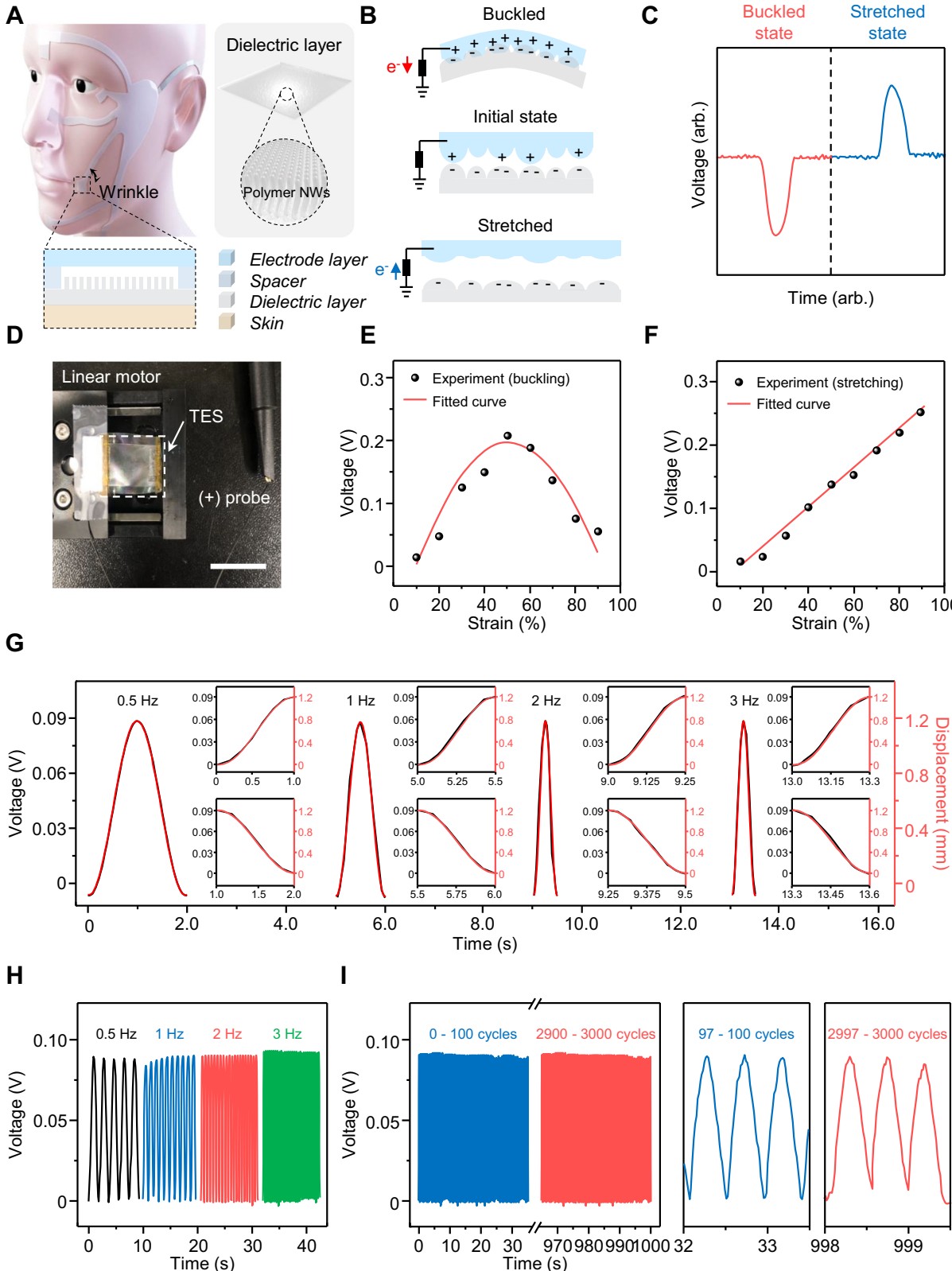

**Fig. 2 | Working mechanism and characterization of the strain sensing unit.**
**A** Schematic illustration of the strain sensing unit. Inset: enlarged view of the sensing unit detecting facial strain. **B** Electrical potential distribution of the strain sensing unit under buckled and stretched state. **C** Output electrical signals of the strain sensing unit during the buckle-stretch cycle. **D** Real image of experimental set-up for output measurements. Scale bar: 1 cm. **E** and **F** Sensitivity measurement during buckling (**E**) and stretching of the sensing unit (**F**). **G** Response time measurement with various frequencies. Insets: enlarged views of the loading and unloading processes in one cycle. **H** Generated voltage signals of the sensing unit with various frequencies at a constant strain of 40%. **I** Mechanical durability test for up to 3000 continuous working cycles and enlarged views of different operation cycles, respectively.

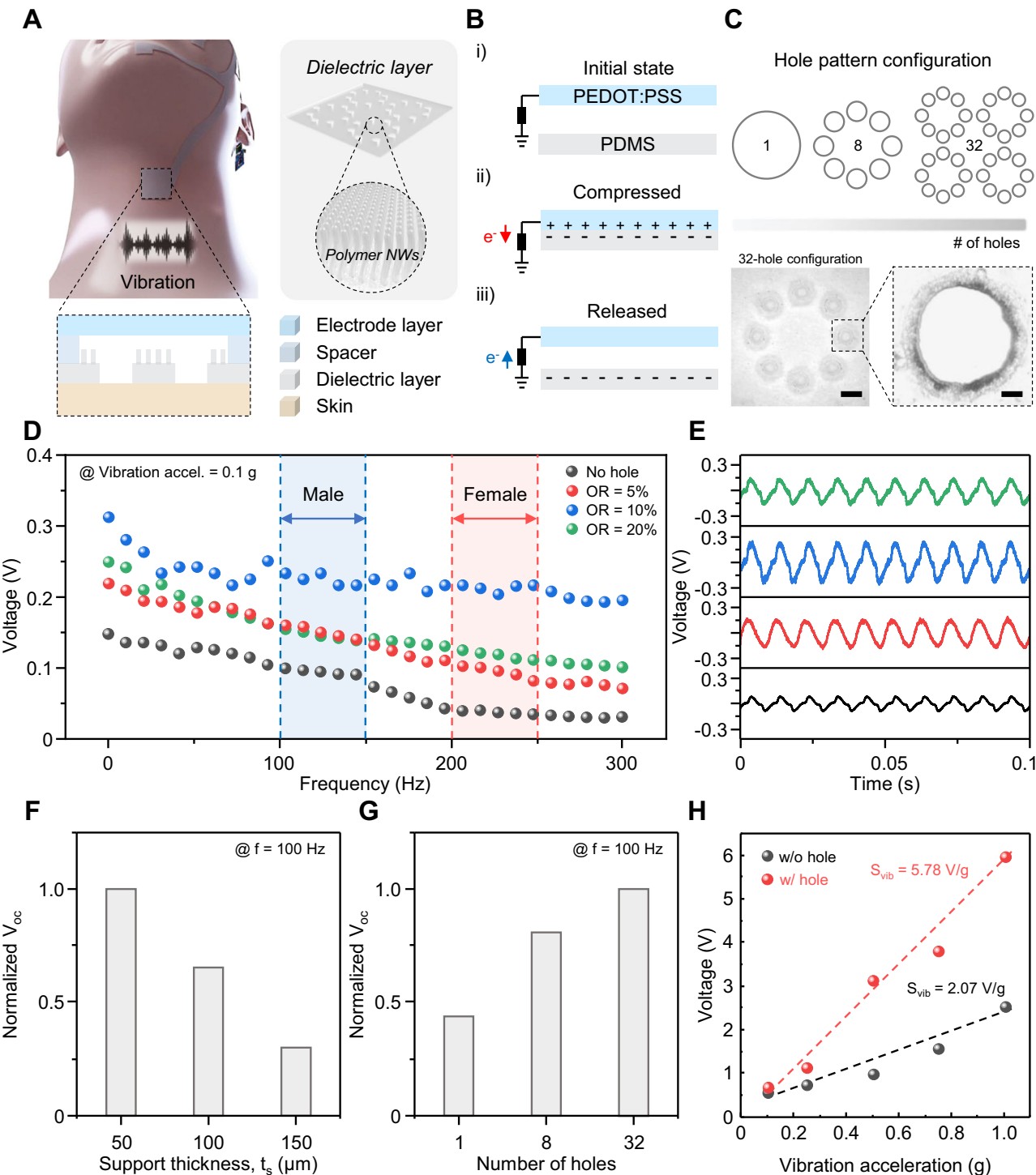

**Fig. 3 | Working mechanism and characterization of the vibration sensing unit.**
**A** Schematic illustration of the vibration sensing unit. Inset: enlarged view of the sensing unit detecting vocal-cord vibration. **B** Electrical potential distribution of the sensing unit during working cycle. **C** Schematic of hole pattern configuration applied in vocal vibration sensor and SEM images of the holes in 32-hole configuration. Scale bar: 2 mm (inset: magnified view showing an acoustic hole. Scale bar: 400 μm). **D** Frequency response data ($V_{oc}$ as a function of acoustic frequency) for the vibration sensing unit with different open ratios (ORs) of 5, 10 and 20. The vocal cord frequency ranges for male and female are colored blue and red, respectively. **E** Measured data plots of output voltage signals per each different OR at the testing frequency of 100 Hz. **F**, **G** Effects of support thickness and number of holes on vibration sensitivity at working frequency of 100 Hz. For each graph, PDMS used as diaphragm material, acoustic holes were patterned on the diaphragm, and the structural parameters were fixed as follows unless otherwise specified: diaphragm thickness of 50 μm, support thickness of 50 μm and an array of 32 holes. The error bars indicate the s.d. of the normalized $V_{oc}$ at the measured frequency of 100 Hz. **H** Comparison of measured output voltage between the vibration sensing unit with and without holes.

## Wireless data processing process and machine learning based real time classification

Figure 4A and B provides real images of the whole PSiFI mask and the participant wearing the PSiFI mask properly laminated onto the participant's face, which made it look transparent and comfortable enough to be worn for long time and communicate well without interrupting expressions that can be caused by a colored device. As schematically depicted in Fig. 4C, our wireless data acquisition and transfer process was carried out from the data collection of the skin-integrated facial mask by the several centimeter size of circuit board as a signal transmitter powered by a tiny portable battery to wirelessly transmitted data received by the main board as the receiver connected to the laptop for storing data used to be datasets for the machined learning.

Figure 4D provides collected triboelectric signal patterns from each modal sensor such as lip, eye, glabella, nose, chin (for strain sensing unit) and vocal cord (for vibration sensing unit). As for the acquired signals from the strain sensing units, distinct patterns were exhibited in accordance with the different facial expressions such as happiness, surprise, disgust, anger and sadness that the participant expressed. As for the signals from the vocal sensing unit, each signals for different speech from the syllable such as "A", "B", "C" to the simple sentence such as "I love you" clearly exhibited its own distinct patterns and were further transformed by fast Fourier transformation (FFT) which converts data from time domain to frequency domain to find remarkable patterns in frequency domain so that the pattern recognition performed well. We conducted separate training for the vocal and strain signals as the interdependence between verbal and nonverbal expressions appears to be relatively insignificant when compared to the distinct and concurrent measurements of the multi-modal inputs (Supplementary Fig. 8).

When it comes to machine learning, we applied the CNN algorithm as an example of algorithm for classification. Specifically, we utilized one-dimensional CNN to classify the facial expressions and two-dimensional CNN for speech classification, respectively (Supplementary Fig. 9 and Table S2). Generally, the more datasets our classifier trains, the better performance it shows. However, it is not viable and time consuming to test the sensor integrated wearable mask to many people in practical terms. The facial muscle movements, vocal cord vibration and sensor values corresponding to the verbal/non-verbal expressions of the new users would be different from those of the previous users since every human has its own characteristics. We therefore need to adapt to a network which can be trained with even small amounts of datasets and tuned with the new datasets from the new users.

Figure 4E provides schematic diagrams showing the overall process from data achieving pre-trained model trained with enhanced accuracy by introducing data augmentation technique (Supplementary Fig. 10 and Table S3) to fine-tunned network for personalization by exploiting pre-trained parameters called as transfer learning, which enables the network to be trained in reduced time and effectively adapt to new user's datasets so that it made the real time classification possible. In detail, a participant repeated, respectively, verbal and non-verbal expression 20 times to demonstrate reliability for a total acquisition of 100 recognition signal patterns per each expression. 70 patterns of total were randomly selected from the acquired signals to serve as the training set which are subsequently augmented 8-fold based on different methods (Jittering, Scaling, Time-warping, Magnitude-warping) for effective learning, and the remaining 30 signals were assigned as the test set. Furthermore, according to the previous report, it was found that the movement and activation patterns of facial muscles during facial expressions was not dissimilar depending on the individuals[62–64]. Based on this fact, we anticipate that the network can get used to adapt to new expressions from new users by rapidly training the corresponding learning data. As for the transfer learning,

after the initial participant had firstly trained with the classifier by the above-mentioned training method, the following participants were wearing with the PSiFI device and able to fast train with the classifier by only repeating 10 times each on both expressions, which successfully allow the real-time classification to be demonstrated. When it comes to practical application, compared with other classification methods based on various kinds of video camera and microphone, our PSiFI mask is free from environmental restrictions such as the location, obstruction, and time. As shown in Fig. 4F, the real-time classification result for combined verbal/nonverbal expressions without any restriction exhibited very high accuracy of 93.3% and even the decent accuracy of 80.0% was achieved despite carrying out the classification with obstruction such as wearing a facial mask (Supplementary Movie 3).

## Digital concierge application in VR environment

As for the application with the PSiFI, we brought in VR environment which allows individuals to experiment with how their emotions could influence and can be expressed and implemented into specific situations in the virtual world[72–74]. This in turn can deepen communications in VR environment by engaging with human emotions. In this sense, we selected digital concierge application that can be enriched with emotional information in terms of practical use and usability. The digital concierge is likely to be anticipated that it can provide user-oriented services which improve quality of user's life by promoting user's experience. Herein, for the first time, we demonstrated the application which offers a digital concierge service operated with our PSiFI based on HMI in VR environment of Unity software as shown in Fig. 5.

Figure 5A provides conceptual schematic showing how human and machine can interact smartly with personalized emotional context by wearing the PSiFI. To realize this, we demonstrate VR-based digital concierge application via HMI with our PSiFI as the overall process was shown in Fig. 5B. Specifically, the digital concierge system was operated based on conversation between the user's avatar and randomly generated avatar who serves as the virtual concierge. Additionally, we built the digital concierge to provide various application services from smart home to entertainment by taking into account the situations which take place very probably in real life.

Figure 5C provides three different scenarios demonstrating smart home, office, and entertainment application in Unity space (Supplementary Movie 4; for details, see the "Methods" secton). As for the first scenario for smart home application, the digital concierge accessed the user's mood of sadness and recommend some playlist from website to relieve the mood despite of user's simple word. As for the second scenario for office application, the digital concierge was able to check if the user understands contents of presentation and pop out new window showing content interpretation that helps to promote user's understanding. As for the last scenario for entertainment application, the digital concierge identifies user's reaction to the movie trailer and curates user-friendly contents in accordance with user's reaction. The applications with our PSiFI-based HMI and built-in VR space can be greatly diversified with learning and adapting new data regarding verbal and non-verbal expressions from new users so that we strongly anticipate our highly personalized PSiFI platform contributes to various practical applications such as education, marketing, and advertisements that can be enriched with emotional information.

## Discussion

In this work, we proposed a machine-learning assisted PSiFI for wearable human emotion recognition system. The PSiFI was made of PDMS-based dielectric and stretchable conductor layers that are highly transparent and comfortable as possible to wear in real life. By endowing our PSiFI with multi-modality to detect simultaneously both facial and vocal expressions using self-powered triboelectric-based sensing units, we can acquire better emotional information regardless

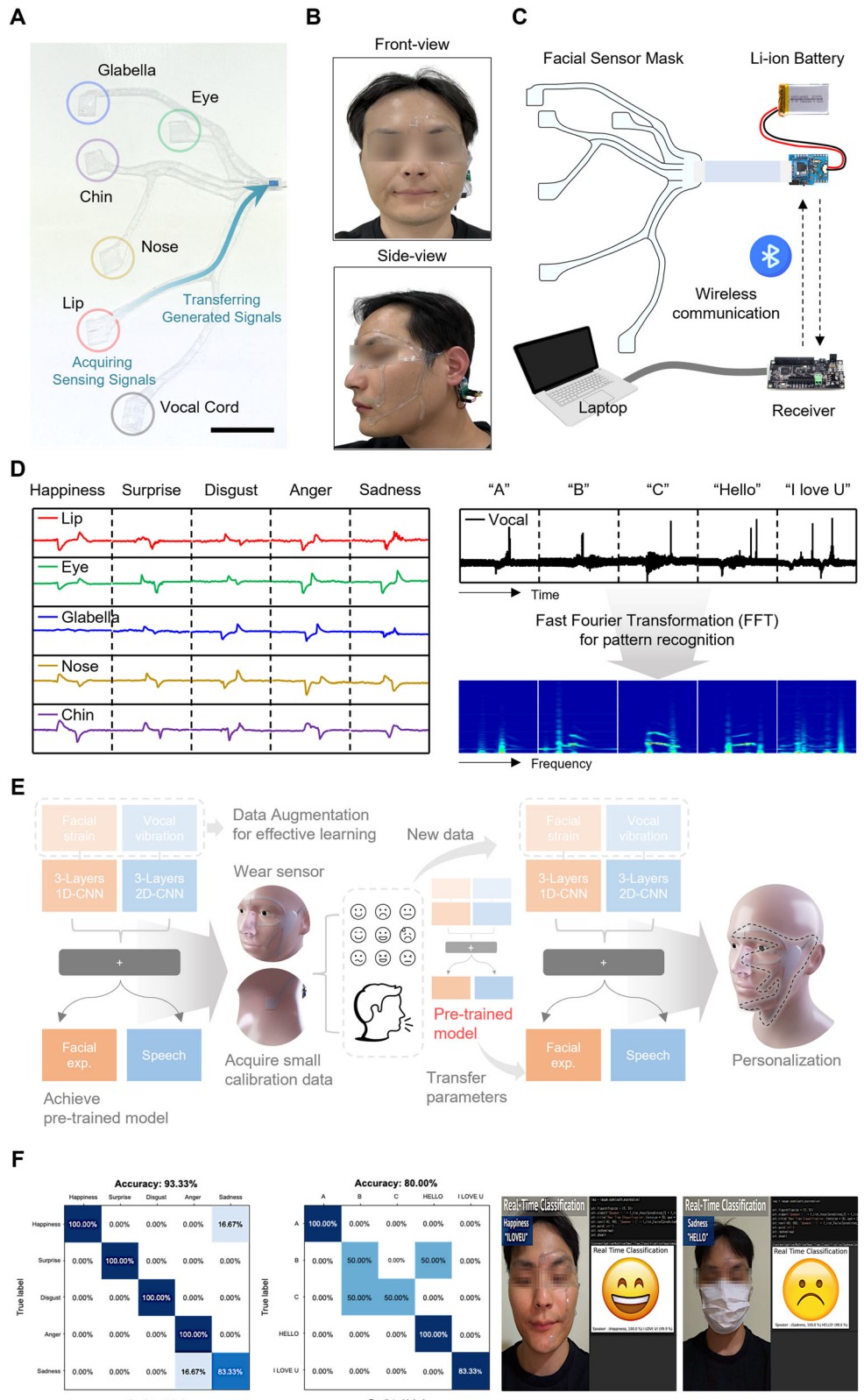

**Fig. 4 | Real-time emotional speech acquisition. A** Photograph showing the multimodality of the PSiFI attached to active units such as glabellar, eye, nose, lip, chin, and vocal cord for simultaneous verbal/non-verbal data collection. Scale bar: 2 cm. **B** Real images of front (top) and side view (bottom) of the participant wearing the PSiFI. **C** Schematic diagrams of the wireless emotional speech classifying system including PSiFI, signal processing board for wireless data transfer. **D** Facial strain and vocal vibration signals were collected from the skin-integrated interface. **E** The processes of learning algorithm architecture implemented in our classification system where machine learning methods such as data augmentation and transfer learning were applied to efficiently reduce training time for the real-time classification. **F** Comparison of confusion matrix (left) and captured images (right) in real-time classification between without and with an obstacle such as a mask.

**A**

Human-Machine Interaction with Personalized Emotional Context

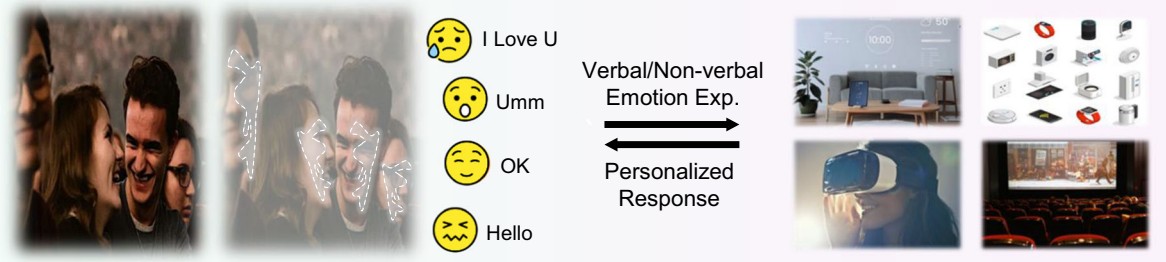

Wear User's PSiFI                    Smart Interaction with Machines using Personalized Emotional Context

**B**

Emotional Speech-based VR Application

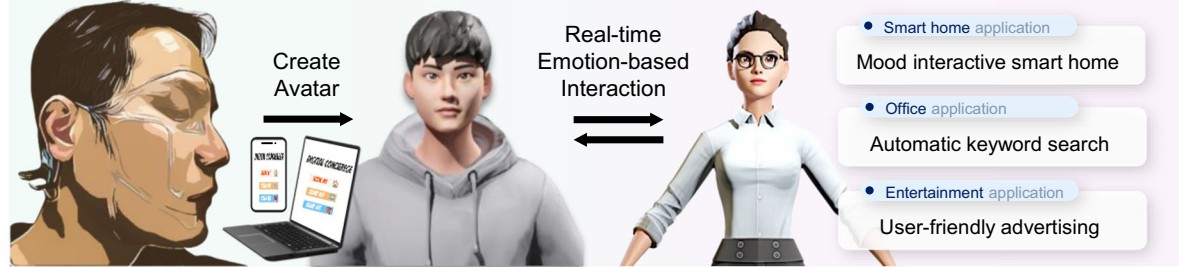

User's Avatar with Personalized Emotion                    Digital Concierge with Optimized Tasks

**C**

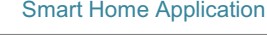

| Scenario #1 | Scenario #2 | Scenario #3 |
|---|---|---|
| Smart Home Application | Office Application | Entertainment Application |

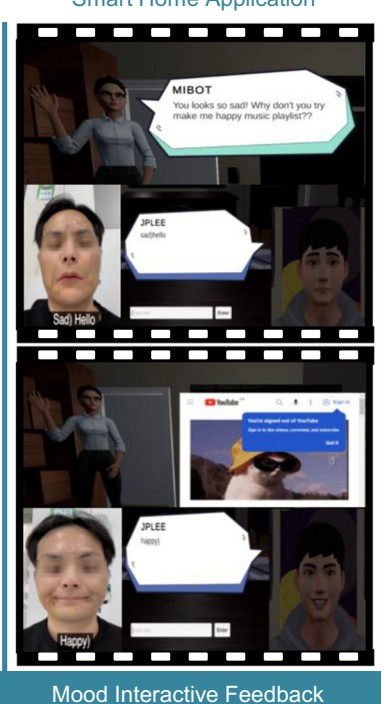

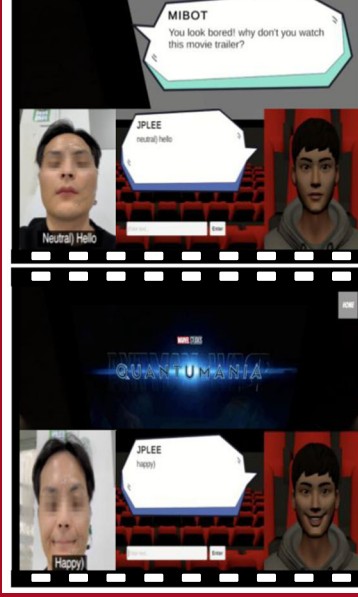

| Mood Interactive Feedback | Automatic Keyword Search | User-friendly Advertising |

**Fig. 5 | The demonstration for digital concierge based on the emotional speech classifying system in VR environment. A** Conceptual illustration of human machine interaction with personalized emotional context achieved by wearing user's PSiFI. **B** Schematic diagram of the way the user interacts with the digital concierge providing various helpful services. **C** The corresponding captured images of three different scenarios as tasks (such as mood interactive feedback, automatic keyword search and user-friendly advertising) of digital concierge which likely take place in various places such as home, office and theater in VR environment of Unity software.

of external factors such as time, place, and obstacles. Furthermore, we realized wireless data communication for real-time human emotion recognition with the help of designed data-processing circuit unit and the rapid adapting learning model and achieved acceptable standard in terms of test accuracy even with the barrier such as mask. Finally, we first demonstrated digital concierge application in VR environment capable of responding to user's intention based on the user's emotional speech information. We believe that the PSiFI could assist and accelerate the active usage of emotions for digital transformation in the near future.

## Methods

### Materials
PDMS was purchased from Dow corning which consists of elastomer base and curing set (10:1 wt/wt). Aqueous dispersions of PEDOT:PSS solution (>3%), ethylene glycol (99.8%), and Au nanoparticles (Au NPs) (~100 nm) dispersion in deionized water (DI) was purchased from Sigma-Aldrich. Acetone (99.5%) and isopropyl alcohol (IPA) (99.5%) were purchased from Samchun Chemical.

### Preparation of conductive dispersion and stretchable conductor
An aqueous solution of PEDOT:PSS was firstly filtered through a 0.45 mm nylon syringe filter. Next, 5 wt% DMSO was added to the solution, and it was then mixed with 50 wt% IPA solvent by vigorously stirring at room temperature for half an hour. Subsequently, the base monomer and curing agent were mixed with a weight ratio of 10:1 at room temperature and then, placed into the vacuum desiccator to degas the PDMS mixture. After 40 min, 1 mL of mixture was spread in the form of a continuous layer onto the cleaned Kapton film as a substrate using a micrometer adjustable film applicator, and allowed to solidify into an amorphous free-standing film by heating on an oven at 90 °C for 5 min. The prepared conductive dispersion was subsequently coated on the PDMS to anchor the conductive polymers within the PDMS matrix before the film is fully solidified.

### Fabrication of nanowire-based surface modification of dielectric film
Nanowires on the surface of the PDMS film were formed by using inductively coupled plasma (ICP) reactive ion etching. The dielectric films with a thickness of 50 μm were first cleaned subsequently by Acetone, IPA and DI, then blown dry with nitrogen gas. In the etching process, Au NPs were prepared by vortex mixer for homogeneous distribution and deposited by drop-casting. After 30 min of drying in oven at 80 °C, the Au NPs were coated on the dielectric surface as a nano-patterned mask. Subsequently, a mixed gas including Ar, $O_2$, and $CF_4$ was introduced in the ICP chamber, with a corresponding flow rate of 15.0, 10.0, and 30.0 sccm, respectively. The dielectric films were etched for 300 s to obtain a nanowire structure on the surface. One power source of 400 W was used to yield a large density of plasma, while another 100 W was used to accelerate the plasma ions.

### Fabrication of hole-patterned dielectric films
Arrays of circular acoustic holes with various shapes and distributions were fabricated and punched through the PDMS film (thickness 100 μm) using laser-cutting technology (Universal Laser Systems Inc.). The diameter of the smallest hole is 500 μm, which is close to the line-width limitation of the laser cutting on a plate surface.

### Fabrication of self-powered sensing units
As for the strain sensing unit, the prepared stretchable conductor was cut in the size of 1 cm × 1 cm. Next, a flat flexible cable (FFC) was attached with the double-sided medical silicone tape (3M 2476P, 3M Co., Ltd) for electrical connection (Supplementary Fig. 11). Then, the surface modified dielectric film (thickness 50 μm) was subsequently placed on the layer and used as space-charge carrying layer.

As for the vibration sensing unit, the prepared stretchable conductor was cut in the size of 1 cm × 1 cm. Next, the FFC was attached with the double-sided medical tape for electrical connection like in the strain sensing unit. Then, the 50 μm-thick surface modified and hole patterned PDMS film as dielectric layer was sequentially applied on the layer and used as diaphragm deflecting with the vocal vibration.

### Characterization and measurement
The morphologies and thickness of the PEDOT:PSS embedded stretchable conductor and the nano-patterned dielectrics were investigated by using a Nano 230 field-emission scanning electron microscope (FEI, USA) at an accelerating voltage of 10 kV. Optical transmission measurements of the stretchable conductors were performed on ultraviolet–visible spectrophotometer (Cary 5000, Agilent) from 400 to 800 nm. The sheet resistances ($R$s) of the stretchable conductors were measured using the four-point van der Pauw method with collinear probes (0.5 cm spacing) connected to a four-point probing system (CMT2000N, AIT). For the electrical measurement of the strain sensor unit, an external shear force was applied by a commercial linear mechanical motor (X-LSM 100b, Zaber Technologies) and a programmable electrometer (Keithley model 6514) was used to measure the open-circuit voltage and short-circuit current. For the vibration sensor unit, a Digital Phosphor Oscilloscope (DPO 3052, Tektronix) was used to measure the electrical output signals at the sampling rate of 2.5 GS/s. For the multi-channel sensing system, a DAQ system (PCIe-6351, NI) was used to simultaneously measure electrical output signals of multi-channel sensor units.

### Attachment of the device on the skin
To mount the sensor device completely onto the facial and neck skin, we applied a bio-compatible, ultrathin, and transparent medical tape (Tegaderm TM Film 1622W, 3M) over the edge of the sensor and the metal lines connected to the interface circuit. The medical tape is developed and widely utilized for skin-friendly adhesive solution. Therefore, there was no skin irritation or itch during several hours of wearing. The test was exempted from IRB in accordance with the approval by UNIST IRB Committee. The authors affirm that human research participants provided informed consent prior to inclusion in this study and for publication of the images in Figs. 4 and 5.

### Machine learning for emotion recognition
For the pre-training, a total acquisition of 100 recognition signal patterns per each expression were collected from a participant repeating 20 times each on both verbal and non-verbal expressions, respectively. 70 patterns of total were randomly selected as training set, further augmented 8-fold based on different augmentation methods (Jittering, Scaling, Time-warping, Magnitude-warping), and the remaining 30 signals were assigned as the test set. After pre-processing step for the datasets such as trimming in accordance with input size of the neural network and converting to image by FFT, the 1D-CNN and 2D-CNN were applied for non-verbal expression and verbal-expression training. With this pre-trained classifier, a new user can rapidly customize the classifier with its own data by repeating 10 times each on both expressions, known as transfer learning, the real-time classification was successfully demonstrated.

### Demonstration of the application
The three-dimensional (3D) VR environment that the user saw was provided by Unity3D on a computer, the facial strain and vocal vibration sensing data were sent to Unity3D through wireless serial communication from Buleinno2, and the interaction between PSiFI and the computer was done by PySerial package in python. We built VR-based digital concierge scenario comprising of environmental assets and generated avatars as follows. The virtual environments assets such as home, office, and theater were downloaded at Unity Asset Store. The

avatars used in the VR environments were simply created from individual photo using readyplayer.me website. In demonstration, the generated avatar proceeded the scenario based on the real-time information transmitted from PSiFI and got adaptive responses from the avatar called MIBOT virtually created for digital concierge.

## Reporting summary

Further information on research design is available in the Nature Portfolio Reporting Summary linked to this article.

## Data availability

The data that support the plots within this paper and other finding of the study are present in the paper and/or the Supplementary Information. The original datasets for human emotion recognition are available from https://github.com/MATTER-INTEL-LAB/PSIFI.git.

## Code availability

All codes used for implementation of the data augmentation and classification are available from https://github.com/MATTER-INTEL-LAB/PSIFI.git.

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

## Acknowledgements

This work was supported by National Research Foundation of Korea (NRF) grants funded by the Korean government, NRF-2020R1A2C2102842, NRF-2021R1A4A3033149, NRF-RS-2023-00302525, the Fundamental Research Program of the Korea Institute of Material Science, PNK7630 and Korea Institute for Advancement of Technology (KIAT) grant funded by the Korea Government (MOTIE) (P0023703, HRD Program for Industrial Innovation).

## Author contributions

J.P.L. carried out and designed most of the experimental work and data analysis. H.J., H.S., and S.L. assisted in the materials processing and device fabrication. Y.J. assisted in the machine learning algorithms and analysis of the results. P.S.L. revised and improved the manuscript with technical comments. J.K. supervised the whole project and was the lead contact. All authors discussed and wrote and commented on the manuscript.

## Competing interests

The authors declare no competing interests.
