## [Peer Review File · Nature Communications]

REVIEWER COMMENTS

Reviewer #1 (Remarks to the Author):

Lee et. al. reported a multi-modal human emotion recognition system (authors named it personalized skin-integrated facial interface (PSiFI)) which can efficiently utilize comprehensive emotional information by combining verbal and non-verbal expression data. Authors fabricated a bidirectional triboelectric strain and vibration sensor which is self-powered, stretchable, and transparent. Additionally, authors demonstrated a fully integrated system combining data processing circuit with wireless data transfer that allows them real-time emotion recognition. Finally, they demonstrated digital concierge application in VR environment using machine learning algorithm. The work is quite interesting and is indeed a new take on bidirectional functionality in triboelectric sensor. There are however several major comments/questions that need to be addressed.

1. Fabrication process of nanostructure formation on the dielectric PDMS surface is not clear. While using Au NP mask for nanowire patterning is a common technique used for Si nanowire fabrication, authors did not describe how they achieved homogeneously distributed Au nanoparticles on PDMS surface? How were Au nanoparticles bonded to PDMS? And finally, after RIE, how they remove Au nanoparticle masks. Detailed description of the method is expected.
2. Working mechanism of bidirectional triboelectric sensor is not fully clear. If a signal is detected from facial strain during different emotional condition, how to differentiate the signals whether it is originated from buckling motion or strain motion? Additionally, from fig. 2E, both 30% strain and 70% strain make same voltage output. How can it be differentiated?
3. Authors should provide enlarged version of signal as insets for cyclic test (fig. 2I) to see how shape of signals changes after many cycles.
4. Definition of open ratio (OR) should be clear. May be using percentage of opening area can be clearer to readers.
5. How simulation correlates experimental outcome? In simulation generated voltage in kV whereas experimental output voltage in volts (or tens of mVs)? In addition, details parameter for simulation needs to be provided.
6. Author claimed, "On the other hand, the larger number of diaphragms with the same OR condition makes the diaphragms deflect more vigorously, thus enhancing the triboelectric output performance." – How larger number of diaphragms help deflect more? Should not the larger number of diaphragms increase the thickness of device, higher stiffness, hence lower vibration?
7. How did authors wire the device (TES) to data acquisition unit? How reliable of long wiring on soft PDMS from device to DAQ?

Reviewer #2 (Remarks to the Author):

The authors proposed a human emotion recognition system utilizing flexible sensors and convolutional neural network techniques. This is a comprehensive engineering work and the story is mostly well told. Please address my following concerns before considering accepting the manuscript.

1) Line 67-69, "... imposes constraints on the range of applications". This is unclear what kind of application is not achievable by previous approaches.

Line 70, "... using single modal data, thus limiting the use of higher-level and comprehensive emotional contexts" It's highly recommended that comparison between approaches using single-modal data and multi-modal data.

2) Figure 2. The response of strain sensors is poorly presented. Detailed descriptions such as the cycled tension test and the response under different strain rates should be shown.

Reviewer #3 (Remarks to the Author):

The work reports the classification of emotional expression using the personalized skin-integrated facial interface (PSiFI) system based on triboelectric strain and vibration sensors. Even though the novelty in materials and devices are not high, the work is interesting in their approach using the physical sensors and real-time classification of facial and vocal expressions using wearable signal transfer and machine learning. There are some unclear presentations of the data and lack of details in the methods and explanation. The paper needs a major revision.

1. In Figure 2, captions of e,f are not correctly described and no caption for g.

2. In detection of vocal cord vibration and facial expression, the response characteristics is expected to be different. How will the difference in the sensed data in individuals affect machine learning methods and results? More detailed explanation or example is required (Fig. 4e).

3. Even though the sensors were used for detection of facial and vocal expressions, there is no machine learning based on multi-modal inputs from two expressions for classification of emotion.

4. For vocal cord vibration detection, mechanical vibration on the vocal cord rather than sound pressure change is measured. The role of acoustic holes should be explained based on the vibrational characteristics of the layer with acoustic holes in more clear way.

5. For machine learning (Fig. 4). More detailed description of the generation of data set for training and classification should be described since the data amount affect the accuracy critically. Reliability of data augmentation should be evaluated. For the data set generation, the reproducibility of sensing response to specific stimuli needs to be secured.

Reviewer #4 (Remarks to the Author):

Response to Reviewer #1's Comments

General Comment: Lee et. al. reported a multi-modal human emotion recognition system (authors named it personalized skin-integrated facial interface (PSiFI)) which can efficiently utilize comprehensive emotional information by combining verbal and non-verbal expression data. Authors fabricated a bidirectional triboelectric strain and vibration sensor which is self-powered, stretchable, and transparent. Additionally, authors demonstrated a fully integrated system combining data processing circuit with wireless data transfer that allows them real-time emotion recognition. Finally, they demonstrated digital concierge application in VR environment using machine learning algorithm. The work is quite interesting and is indeed a new take on bidirectional functionality in triboelectric sensor. There are however several major comments/questions that need to be addressed.

Author reply for general comment: We appreciate the reviewer's dedicated time and expertise in conducting a thorough review of our work. All comments are very valuable and insightful for the improvement of our paper. We have carefully addressed each comment with point-by-point replies as follows.

1) Comment: Fabrication process of nanostructure formation on the dielectric PDMS surface is not clear. While using Au NP mask for nanowire patterning is a common technique used for Si nanowire fabrication, authors did not describe how they achieved homogeneously distributed Au nanoparticles on PDMS surface? How were Au nanoparticles bonded to PDMS? And finally, after RIE, how they remove Au nanoparticle masks. Detailed description of the method is expected.

Author reply: Thank you for the reviewer's helpful comment. Basically, we purchased the Au NPs suspension from Sigma-Aldrich and made the Au NPs homogeneous by vortex mixer. After 30 min drying in oven at 80°C, we were able to prepare the Au NPs coated PDMS. As the schematic of the RIE etching process was drawn in Fig. S3, the Au nanoparticles masks were subsequently etched out as well with the controlled power and time in the end based on the reference paper (DOI: 10.1021/jp907072z). We added the details of the fabrication process for the aligned polymer nanowires in the Methods section as follows.

In Page 14,

“... In the etching process, Au NPs were prepared by vortex mixer for homogeneous distribution and deposited by drop-casting. After 30 mins of drying in oven at 80°C, the Au NPs were coated on the dielectric surface as a nano-patterned mask.”

2) Comment: Working mechanism of bidirectional triboelectric sensor is not fully clear. If a signal is detected from facial strain during different emotional condition, how to differentiate the signals whether it is originated from buckling motion or strain motion? Additionally, from fig. 2E, both 30% strain and 70% strain make same voltage output. How can it be differentiated?

Author reply: Thank you for the reviewer’s helpful comment. As shown in Fig.2C, our bidirectional sensor can differentiate whether the signals originated from buckling and strain motion by sensing the sign of measured triboelectric signals. As for the concern about the same voltage with different strains, the facial muscle strain is reported to be normally in the range of 0% to 40% according to one of our references (Ref.40). We can also address the reviewer’s concern by comparing the width of the time between the different strains to differentiate them as shown in Fig. S6 and adding corresponding information in the manuscript as follows.

Fig. S6. Comparison of the output voltage with 30% strain and 70% strain under working frequency of 0.5 Hz.

In Page 7,

→ “The signals in the non-linear region were differentiated with the difference in the width of time as shown in Supplementary Fig. 6.”

3) Comment: Authors should provide enlarged version of signal as insets for cyclic test (fig. 2I) to see how shape of signals changes after many cycles.

Author reply: Thank you for the reviewer’s helpful comment. According to the reviewer’s comment, we added an enlarged version of the signal as insets for the cyclic test in Fig.2i and confirmed that there is no significant change even after many cycles as below.

Fig. 2i, Mechanical durability test for up to 3000 continuous working cycles (left) and enlarged views of different operation cycles (right).

4) Comment: Definition of open ratio (OR) should be clear. Maybe using percentage of opening area can be clearer to readers.

Author reply: Thank you very much for your constructive suggestions for our work. According to the comments, we *modified* the OR from “5, 10, 20” to 5%, 10%, 20% in Fig.3d by formatting as a percentage for better understanding to readers as follows.

Fig. 3d, Response time measurement with various frequencies.

5) Comment: How simulation correlates experimental outcome? In simulation generated voltage in kV whereas experimental output voltage in volts (or tens of mVs)? In addition, details parameter for simulation needs to be provided.

Author reply: Thank you for the reviewer’s helpful comment. According to the reviewer’s comment, we modified the simulated generated voltage in accordance with the real size

parameters in Fig.S6 and added details for the simulation in Table 1 as below. For the sake of simplicity, we simulated and compared the generated output voltage based on the conventional contact-separation mode between the pristine and nanostructured condition, which may result in the difference between experimental and simulation in terms of output performance. For your information, there are reference papers where the simulation results which tend not to accurately show matching with the real results but to show tendency and expectation of the simulated voltage of the TENG device on experimental results based on different relative permittivity and surface charge densities are shown as follows¹⁻⁴. (DOI: 10.1021/acs.nanolett.0c01987, DOI: 10.1002/adma.201402491, DOI: 10.1038/s41467-019-13166-6, DOI: 10.1038/s41467-018-06045-z)

Fig. S6a, Calculated V_{oc} depending on the gap distance.

Fig. S6b, Calculated electrical potential distribution at maximum separated position.

Table S1 | Parameters utilized in the theoretical calculation of the sensors.

Common parameters		
PEDOT:PSS	Length (mm)	10
	Width (mm)	10
	Thickness (mm)	0.1
	$\epsilon_{\text{PEDOT:PSS}}$	1000
Polydimethylsiloxane (PDMS)	Length (mm)	10
	Width (mm)	10
	Thickness (mm)	0.2
PDMS Nanowires (NWs)	ϵ_{PDMS}	2.75
	Diameter (μm)	0.01
	Thickness (μm)	1

Parameters for strain sensor			
Triboelectric surface charge density, σ (nC m ⁻²)	50		
Maximum separation distance, x_{max} (mm)	2		
Parameters for vibration sensor			
Hole diameters (mm) (Open ratio = 20%)	1 hole	8 holes	32 holes
	2.52	0.89	0.45

6) Comment: Author claimed, “On the other hand, the larger number of diaphragms with the same OR condition makes the diaphragms deflect more vigorously, thus enhancing the triboelectric output performance.” – How larger number of diaphragms help deflect more? Should not the larger number of diaphragms increase the thickness of device, higher stiffness, hence lower vibration?

Author reply: Thank you for the reviewer’s helpful comment. First, we are truly sorry for the mistake and for making you confused by miswriting the diaphragms instead of holes. We corrected “diaphragms” into “holes” in Fig.3g and the corresponding part in the manuscript as well. As the holes are increased with same OR condition, the stiffness of diaphragm is decreased and the vibration is generated homogeneously, which help the diaphragm deflect more. The number of holes is limited by the resolution of the laser cutting machine. Additionally, we added the simulated deflection at the center of the diaphragm and deflection distribution of the diaphragm with various number of holes (no hole to 32 holes) in Fig.S6c and d as follows.

Fig. 3f, g, Effects of support thickness and number of **holes** on vibration sensitivity at working frequency of 100 Hz. For each graph, PDMS used as diaphragm material, acoustic holes were

patterned on the diaphragm, and the structural parameters were fixed as follows unless otherwise specified: diaphragm thickness of $50\ \mu\text{m}$, support thickness of $50\ \mu\text{m}$ and an array of 32 holes. The error bars indicate the s.d. of the normalized V_{oc} at the measured frequency of 100 Hz.

Fig. S6c, Calculated deflection at the center of the diaphragm with various number of holes. **Fig. S6d**, Deflection distribution of the diaphragm with various number of holes calculated under input frequency of 100 Hz and input force of 2 mN.

7) Comment: How did authors wire the device (TES) to data acquisition unit? How reliable of long wiring on soft PDMS from device to DAQ?

Author reply: Thank you for the reviewer’s helpful comment. As we noted in the Methods section, the device (TES) was wired to data acquisition unit with flat flexible cable (FFC) by double-sided medical silicone tape. We selected the FFC for electrical connection due to high insulation without possible migration and connection stability even when the number of conductors is increased. For the stability of our device, we could say that it can retain connection stability over several hundred cycles per single device. The enlarged view of the photograph of the region for wiring and schematic images for the connection are shown in the Fig.S11 as follows.

Fig. S11, (a) Enlarged view of the photograph for wiring and (b) schematic images for the electrical connection in cross-section (left) and side (right) view.

Response to Reviewer #2's Comments

General Comment: The authors proposed a human emotion recognition system utilizing flexible sensors and convolutional neural network techniques. This is a comprehensive engineering work, and the story is mostly well told. Please address my following concerns before considering accepting the manuscript.

Author reply for general comment: We appreciate the reviewer's dedicated time and expertise in conducting a thorough review of our work. All comments are very valuable and insightful for the improvement of our paper. We have carefully addressed each comment with point-by-point replies as follows.

1) Comment: Line 67-69, "... imposes constraints on the range of applications". This is unclear what kind of application is not achievable by previous approaches. Line 70, "... using single modal data, thus limiting the use of higher-level and comprehensive emotional contexts" It's highly recommended that comparison between approaches using single-modal data and multi-modal data.

Author reply: We strongly appreciate reviewer's valuable and helpful comment. To avoid confusion and make it clearer, we elaborated the points further as follows.

Line 67-69

→ "... only limited to one-to-one correlation that imposes constraints on the range of applications such as healthcare, VR where complementary information is needed to approximate natural interaction, and user experience can be enhanced by multiple ways of inputs."

Line 70

→ "... using the single-modal data that can have weaknesses in specific context, thus limiting the use of higher-level and comprehensive emotional contexts^{45,48-50,53-56}. On the other hand, to overcome the drawbacks of each modality for a more resilient system, multi-modal emotion recognition was conducted to draw embedded high-level information by using the combined knowledge from all the accessible data sensing⁵⁷⁻⁵⁹."

Table 1 | Comparison between single modal and multi modal system.

Single modal	• single type of data• less computational resources due to simpler processing of single modality• limit of depth and accuracy of understanding• limits interaction to single mode
--------------	--

Multi modal

- multiple types of data
 - more computational resources due to increased complexity of processing multiple modality
 - simultaneous process with multiple modalities, allowing for a richer understanding and improved accuracy
 - enable natural interaction via multiple modes of interactions
 - help to draw embedded high-level information
-

2) Comment: Figure 2. The response of strain sensors is poorly presented. Detailed descriptions such as the cycled tension test and the response under different strain rates should be shown.

Author reply: We deeply appreciate reviewer's valuable and helpful comment. According to the reviewer's comment, we added the cycled tension test and response time measurement results under different strain rates in the Fig.2g-i as below. As shown in the Fig. 2g, we can see the response for our strain sensor well complied with the external strain motion under the different strain rates from 0.5 Hz to 3 Hz. As for the cycled tension test, our strain sensors were tested up to 3000 operation cycles and showed stable output voltages.

Fig. 2g, Response time measurement with various frequencies. Insets: enlarged views of the loading and unloading processes in one cycle. **Fig. 2h**, Generated voltage signals of the sensing unit with various frequencies at a constant strain of 40% **Fig. 2i**, Mechanical durability test for up to 3000 continuous working cycles (left) and enlarged views of different operation cycles (right).

Response to Reviewer #3's Comments

General Comment: The work reports the classification of emotional expression using the personalized skin-integrated facial interface (PSiFI) system based on triboelectric strain and vibration sensors. Even though the novelty in materials and devices are not high, the work is interesting in their approach using the physical sensors and real-time classification of facial and vocal expressions using wearable signal transfer and machine learning. There are some unclear presentations of the data and lack of details in the methods and explanation. The paper needs a major revision.

Author reply for general comment: We appreciate the reviewer's dedicated time and expertise in conducting a thorough review of our work. All comments are very valuable and insightful for the improvement of our paper. We have carefully addressed each comment with point-by-point replies as follows.

1) Comment: In Figure 2, captions of e,f are not correctly described and no caption for g.

Author reply: Thank you for the reviewer's helpful comment. According to the reviewer's comment, we corrected the typo "(d), (e)" into "(e), (f)" and added "g" before the corresponding caption as follows.

In Page 24,

e,f, Sensitivity measurement during buckling (d) and stretching of the sensing unit (e). Response time measurement. Insets: enlarged views of the loading and unloading processes in one cycle

→ e,f, Sensitivity measurement during buckling (e) and stretching of the sensing unit (f). g, Response time measurement. Insets: enlarged views of the loading and unloading processes in one cycle

2) Comment: In detection of vocal cord vibration and facial expression, the response characteristics is expected to be different. How will the difference in the sensed data in individuals affect machine learning methods and results? More detailed explanation or example is required (Fig. 4e).

Author reply: Thank you for the reviewer's helpful comment. According to the reviewer's comment, we elaborated machine learning assisted real-time recognition part from the data collection to machine learning application by giving corresponding information as follows.

In Page 11,

→ "... In detail, a participant repeated respectively verbal and non-verbal expression 20 times to demonstrate reliability for a total acquisition of 100 recognition signal patterns per each expression. 70 patterns of total were randomly selected from the acquired signals to serve as

the training set which are subsequently augmented 8-fold based on different methods (Jittering, Scaling, Time-warping, Magnitude-warping) for effective learning, and the remaining 30 signals were assigned as the test set. Furthermore, according to the previous report, it was found that the movement and activation patterns of facial muscles during facial expressions was not dissimilar depending on the individuals⁶²⁻⁶⁴. Based on this fact, we anticipate that the network can get used to adapt to new expressions from new users by rapidly training the corresponding learning data. As for the transfer learning, after the initial participant had firstly trained with the classifier by the above-mentioned training method, the following participants were wearing with the PSiFI device and able to fast train with the classifier by only repeating 10 times each on both expressions, which successfully allow the real-time classification to be demonstrated.”

In Page 17,

→ **“Machine learning for emotion recognition.** For the pre-training, a total acquisition of 100 recognition signal patterns per each expression were collected from a participant repeating 20 times each on both verbal and non-verbal expressions, respectively. 70 patterns of total were randomly selected as training set, further augmented 8-fold based on different augmentation methods (Jittering, Scaling, Time-warping, Magnitude-warping), and the remaining 30 signals were assigned as the test set. After pre-processing step for the datasets such as trimming in accordance with input size of the neural network and converting to image by FFT, the 1D-CNN and 2D-CNN were applied for non-verbal expression and verbal-expression training. With this pre-trained classifier, a new user can rapidly customize the classifier with its own data by repeating 10 times each on both expressions, known as transfer learning, the real-time classification was successfully demonstrated.”

3) Comment: Even though the sensors were used for detection of facial and vocal expressions, there is no machine learning based on multi-modal inputs from two expressions for classification of emotion.

Author reply: Thank you for the reviewer’s helpful comment. Basically, as shown in Fig.4e, our machine learning is composed of two parallel models of 1d CNN for facial expressions and 2d CNN for vocal expressions respectively. We intentionally decided to use the separated learning models since we thought it could offer us more combinations of emotion recognition, which means we can more manifoldly recognize our emotions so that we could differentiate the feigned emotions by simultaneously considering the results from the multi-modal inputs. Although the multi-modal inputs are affected by each other, as shown in below as Figure 3, the effect was not as major as the trend of the signal patterns could be affected. Moreover, we believe that the effect could be getting insignificant through the learning and data augmentation process, so we think training with separate models is more effective compared to training with a combined model.

Fig. S8. Comparison of the output voltage signals from multi-channel sensors under various facial expressions (i-happiness, ii-surprise, iii-disgust, vi-anger, v-sadness) when they were measured separately or simultaneously with vocal expressions.

In Page 11,

→ “We conducted separate training for the vocal and strain signals as the interdependence between verbal and non-verbal expressions appears to be relatively insignificant when compared to the distinct and concurrent measurements of the multi-modal inputs (Supplementary Fig. 8).”

4) Comment: For vocal cord vibration detection, mechanical vibration on the vocal cord rather than sound pressure change is measured. The role of acoustic holes should be explained based on the vibrational characteristics of the layer with acoustic holes in more clear way.

Author reply: Thank you for the reviewer’s helpful comment. According to the reviewer’s comment, we elaborated the role of acoustic holes in manuscript as follows.

In page 8,

→ “The holes were introduced into the vocal sensing unit as acoustic holes which not only act

as communicating vessels to ventilate an air between two contact surfaces to the ambient air, which results in enhanced flat frequency response but also reduce the stiffness by improving the movement of the rim of diaphragms.”

5) Comment: For machine learning (Fig. 4). More detailed description of the generation of data set for training and classification should be described since the data amount affect the accuracy critically. Reliability of data augmentation should be evaluated. For the data set generation, the reproducibility of sensing response to specific stimuli needs to be secured.

Author reply: Thank you for the reviewer’s helpful comment. According to the reviewer’s comment, we added the detailed information of the dataset involving the data amount for training and classification in the Supplementary information. We also added the description about the data set for training and classification in the Method section in the main manuscript as follows.

Table S3 | Information of the dataset from PSIFI utilized in emotion recognition.

Pre-training					
Dataset	Train set		Test set	Total	
	original	augmented		original	augmented
Facial expression (non-verbal)	70	560	30	100	590
Vocal speech (verbal)	70	560	30	100	590
Transfer-learning					
Dataset	Train set		Test set	Total	
	original	augmented		original	augmented
Facial expression (non-verbal)	35	270	15	50	285
Vocal speech (verbal)	35	270	15	50	285

In Page 14,

→ “... In detail, a participant repeated respectively verbal and non-verbal expression 20 times to demonstrate reliability for a total acquisition of 100 recognition signal patterns per each expression. 70 patterns of total were randomly selected from the acquired signals to serve as the training set which are subsequently augmented 8-fold based on different methods (Jittering,

Scaling, Time-warping, Magnitude-warping) for effective learning, and the remaining 30 signals were assigned as the test set.”

In Page 17,

→ **“Machine learning for emotion recognition.** For the pre-training, a total acquisition of 100 recognition signal patterns per each expression were collected from a participant repeating 20 times each on both verbal and non-verbal expressions, respectively. 70 patterns of total were randomly selected as training set, further augmented 8-fold based on different augmentation methods (Jittering, Scaling, Time-warping, Magnitude-warping), and the remaining 30 signals were assigned as the test set. After pre-processing step for the datasets such as trimming in accordance with input size of the neural network and converting to image by FFT, the 1D-CNN and 2D-CNN were applied for non-verbal expression and verbal-expression training. With this pre-trained classifier, a new user can rapidly customize the classifier with its own data by repeating 10 times each on both expressions, known as transfer learning, the real-time classification was successfully demonstrated.”

References

1. Jin, L., Xiao, X., Deng, W., Nashalian, A., He, D., Raveendran, V., Yan, C., Su H., Chu X., Yang, T., Li, W., Yang, W. & Chen, J. Manipulating Relative Permittivity for High-Performance Wearable Triboelectric Nanogenerators. *Nano Lett.* **20**, 6404–6411 (2020).
2. Wang, S., Xie, Y., Niu, S., Lin, L., Liu, C., Zhou, Y. S., Wang, Z. L. Maximum Surface Charge Density for Triboelectric Nanogenerators Achieved by Ionized-Air Injection: Methodology and Theoretical Understanding. *Adv.Mater.* **26**, 6720–6728 (2014).
3. Luo, J., Wang, Z., Xu, L., Wang, A. C., Han, K., Jiang, T., Lai, Q., Bai, Y., Tang, W., Fan, F. R. & Wang, Z. L. Flexible and durable wood-based triboelectric nanogenerators for self-powered sensing in athletic big data analytics. *Nat Commun* **10**, 5147 (2019).
4. Cheng, L., Xu, Q., Zheng, Y. et al. A self-improving triboelectric nanogenerator with improved charge density and increased charge accumulation speed. *Nat Commun* **9**, 3773 (2018).

REVIEWERS' COMMENTS

Reviewer #1 (Remarks to the Author):

The authors addressed all my comments and questions. I appreciate the changes that have been made. I believe these new additions made the paper much clear.

Reviewer #2 (Remarks to the Author):

All my comments are addressed in a satisfying manner. I suggest accepting the manuscript as is.

Reviewer #3 (Remarks to the Author):

The comments were addressed with addition explanation. The paper is recommended for publication.

Reviewer #4 (Remarks to the Author):
